# UV-C Irradiation and Essential-Oils-Based Product as Tools to Reduce Biodeteriorates on the Wall Paints of the Archeological Site of Baia (Italy)

Paola Cennamo [1,*], Roberta Scielzo [1], Massimo Rippa [2], Giorgio Trojsi [1], Simona Carfagna [3] and Elena Chianese [4]

1   Department of Humanities, University of Naples Suor Orsola Benincasa, 80132 Naples, Italy
2   Institute of Applied Sciences and Intelligent Systems "Eduardo Caianiello", CNR, 80078 Pozzuoli, Naples, Italy
3   Department of Biology, University of Naples Federico II, 80126 Naples, Italy
4   Department of Science and Technology, Parthenope, University of Naples, 80143 Naples, Italy
*   Correspondence: paola.cennamo@unisob.na.it

**Abstract:** This study is aimed to compare, through laboratory experimentations, the efficiency of UV-C irradiation and an essential-oils-based product as tools to reduce the biofilm identified in a semi-hypogeum room located in the archaeological park of Baia, Italy. During this study, the autotrophic component of the original biofilm, mostly composed of Chlorophyceae and Cyanophycean, was isolated in the laboratory, while simultaneously, the composition of the pigments used for the fresco paintings was examined in situ through X-ray fluorescence. These examinations were necessary for the creation of test samples that were similar to the original surfaces and used for subsequent experiments. The plaster testers were contaminated with artificial biofilm, exposed to UV-C at a distance of 80 cm for a fixed time interval and treated with *ESSENZIO©*, a product based on oregano and thyme essential oils, to eradicate the biological species. The treatment's effectiveness was then assessed by employing optical microscopy and spectrometric techniques applied to the areas previously occupied by the biofilm on the different test samples. To obtain an additional parameter to evaluate the treatments efficacy, the concentrations of the photosynthetic pigments were also measured by spectrophotometry. Results showed that biofilms were successfully removed by the irradiation of the surfaces and by the essential-oils-based product at a dilution of 50% in demineralized water with a time of application of 1 h and 30 min; in addition, no visible change of the pigments used on the testers were observed, demonstrating the high efficiency of the treatments against biodeteriogens. The two methods and their different mechanisms of action have provided interesting aspects that suggest a combined strategy to contrast and prevent biological growth in archaeological contexts.

**Keywords:** biodeterioration; essential oils; UV-C irradiation; biofilms; conservation; microbial growth control

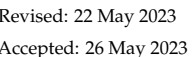

## 1. Introduction

Among the most common conservation problems for historical and artistic monuments, biodeterioration represents one of the most important causes of decorated surface damage. Archaeological areas are highly influenced by the environmental context, which is one of the main factors that lead to the deterioration of artifacts. These peculiar environmental conditions, together with high humidity levels and slight fluctuation in temperatures, allow for the proliferation of heterogeneous biological species typical of semi-confined areas, which contribute to the biodeterioration phenomena [1–3]. The diffusion of biological species over painted surfaces leads to the chromatic variation and degradation of the artifacts [3,4]. The progress in the treatment of biological patinas has motivated the

experimentation of alternative methods (e.g., essential oils, UV-C irradiation, laser cleaning) to control the proliferation of biofilms on historical and artistic objects. The most common biocides in conservative procedures, such as Biotin T, that are characterized by proven effectiveness for the treatment of biodeteriogens are very harmful to the environment and the operators. The most recent experimentations have shown the efficacy of UV-C radiation and essential oil-based products to control the biological growth of different typologies of artifacts [4,5].

UV-C irradiation has been used to treat stone monuments since the 1960s, as reported in Borderie et al. [4] and Liverani et al. [6]. Recently, this method was also tested by Cennamo et al. to create a pilot system to prevent biological growth on the painted surfaces of the tomb ES-07 in Porta Nocera necropolis, located in the Archeological Park of Pompeii [7].

Recently, essential oils were tested in different contexts to remove the biodeteriogens from different kinds of artifacts; as an example, Devreux et al. used these substances for the treatment of stone statues located in the Vatican Gardens as a pilot experiment for new methods aimed to control biological growth on artistic artifacts [8,9].

The Archeological Park of Baia, with its characteristic environmental conditions— high humidity and constant temperatures due, in particular, to the presence of underground thermal water—represents the perfect context for the formation of heterotrophic and autotrophic biofilm. For this reason, the individuation of effective methods to remove biodeteriogens and to prevent their future growth represents a fundamental topic. The area selected to collect and study biofilm was room SB-E0-R07, located in the north area of the Mercury Sector in the Archeological Park of Baia, where a thick layer of biological species was present all over the painted surfaces. A pilot system was then developed to irradiate some pigmented samples with UV-C rays, and other testers were treated with an essential-oil-based product to evaluate and to compare the efficiency of both methods in the removal of the biofilm.

## 2. Materials and Methods

Room SB-E0-R07 is situated in the north zone of the Mercury Sector, in the Archeological Park of Baia (Figures 1 and 2). This area is characterized by the presence of underground salty thermal water, which is one of the main factors that contributes to the high level of humidity recorded here.

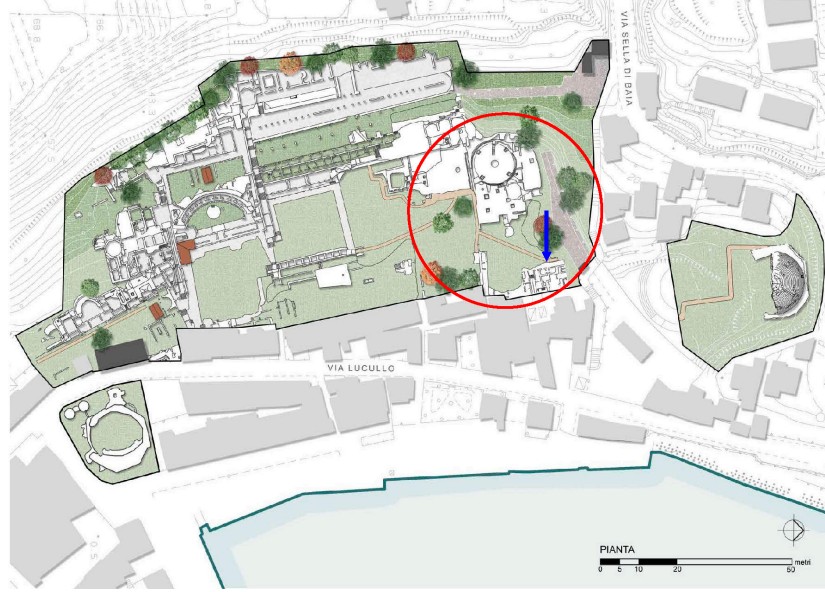

**Figure 1.** Planimetry of the Archeological Park of Baia. The red circle highlights the Mercury Sector, while the blue arrow indicates the SB-E0-R07 room.

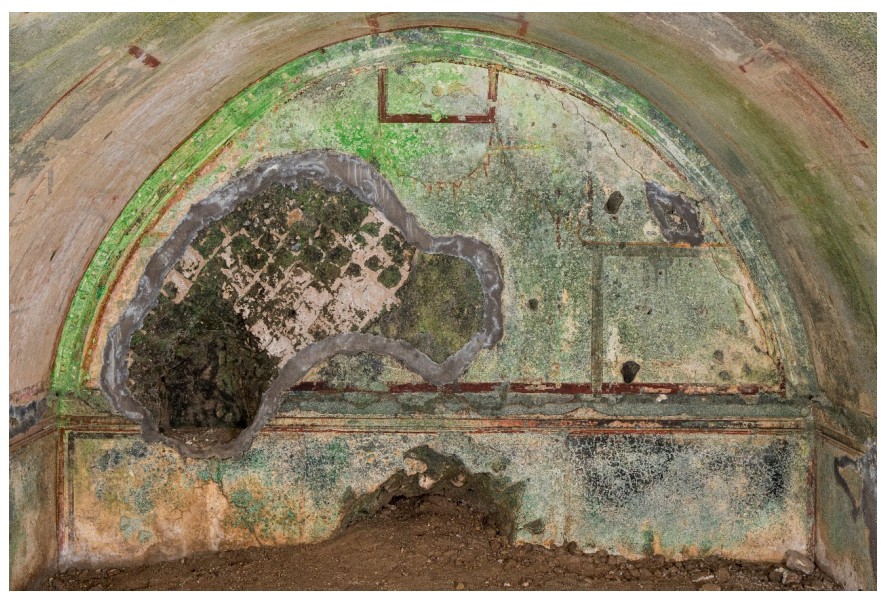

**Figure 2.** Biofilms growing on the wall paintings of the SB-E0-R07 room.

In particular, in the chamber object of the present study, we investigated the variations in the thermohygrometric parameters for one year with the support of four dataloggers (*ORIA©*). After the monitoring period, data were processed to detect any variation in humidity and temperature; as a result, we observed a high value for the average humidity (93%) and a mild value for the average temperature (in the range of 12–14 °C) (Figures 3 and 4).

These factors are the main causes of the proliferation of the biodeteriogens on the fresco paintings in this site; the identification of microbial communities was previously carried out [7,8], showing the presence of different autotrophic and heterotrophic microorganisms, in particular, Cyanobacteria and algae.

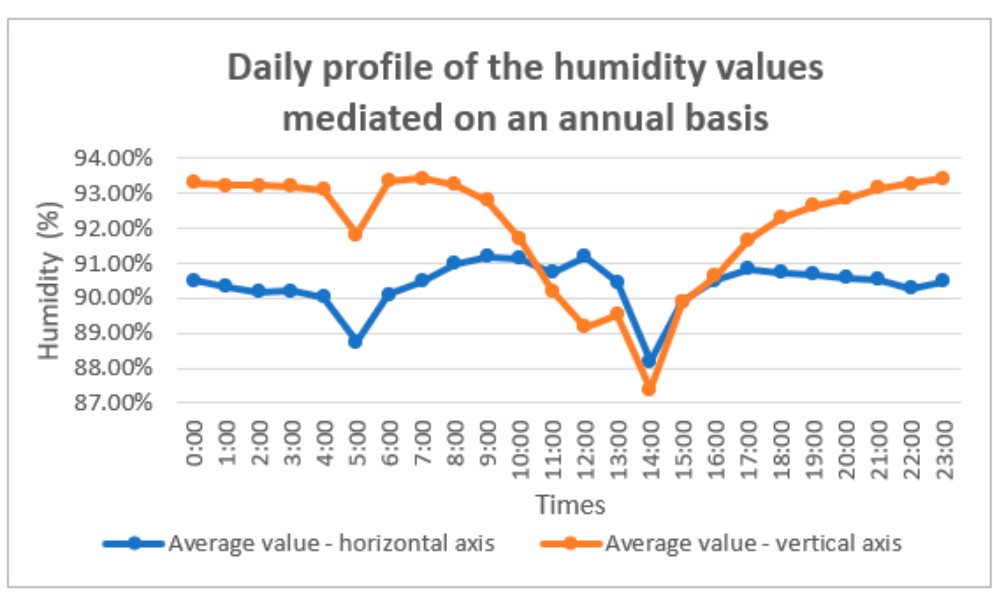

**Figure 3.** Variations in the daily profile of the humidity of room SB-E0-R07 on an annual basis.

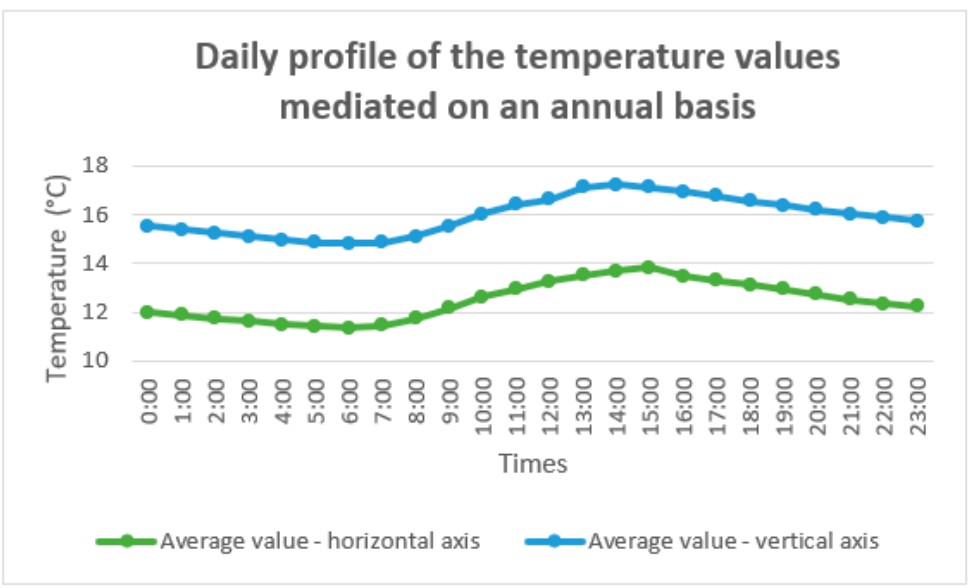

**Figure 4.** Variations in the daily profile of the temperature of room SB-E0-R07 on an annual basis.

### 2.1. Chemical-Physical Analysis (XRF)

X-ray fluorescence analysis Assing, Rome, Italy) were performed to detect the elementary composition of the original pigments of the wall paintings. Red, yellow, white and black colours, in addition to the composition investigation of the original plaster, were analysed with a portable XRF-Q Assing spectrometer (Assing, Rome, Italy), a tungsten tube, and a PiN silicon diode detector with beryllium window, at operating conditions of 30 KV and 0.5 mA (counting time: 30 s).

### 2.2. Cultivation of Algae

Following the detection of microalgae belonging to Chlorophyceae and Cyanophycean, two different growth media, BG11 and BBM [10,11], were used to determine growth in the laboratory. In order to obtain exponential growth, the cultures were stored in a climatic chamber at 37 °C for 12 days. Artificial biofilm preparation: A series of test samples were artificially reproduced using the same construction materials present in the nymphaeum and pigments used for the original surfaces (Figure 1). A total of two test samples, four for each of the three original pigments, were prepared and inoculated with the isolated cultures in the laboratory until they were completely covered with biofilm; in Figures 5 and 6 are shown (a) black samples; (b) yellow samples; (c) white samples and red samples (d), following the procedure of Cennamo et al. [10]. The biofilm growth on the substrate was monitored in the following days.

### 2.3. Essential Oils-Based Product Treatments

The samples were treated with *ESSENZIO©,* a product based on thyme and oregano essential oils and produced by *IBIX©*, applied by brush and diluted in demineralized water as suggested by the procedure of Devreux et al. [9], at different concentrations (10%, 20%, 50%, and pure), and it was removed with demineralized water after different durations of action (from 30 to 90 min) (Figure 5).

### 2.4. UV-C Treatments

The test samples were exposed to radiation using two UV-C lamps (Fluorescent Compact G9W Lynx, Sylvania, OA, USA, 2 × 9 W each = 18 W total, λ max = 254 nm). A total of 5 mL of algal culture was extracted from the test samples to be used as a control. The irradiations were conducted at a distance of 80 cm for a duration of 8 h a day every day, for a total duration of exposure of 24 h (Figure 5).

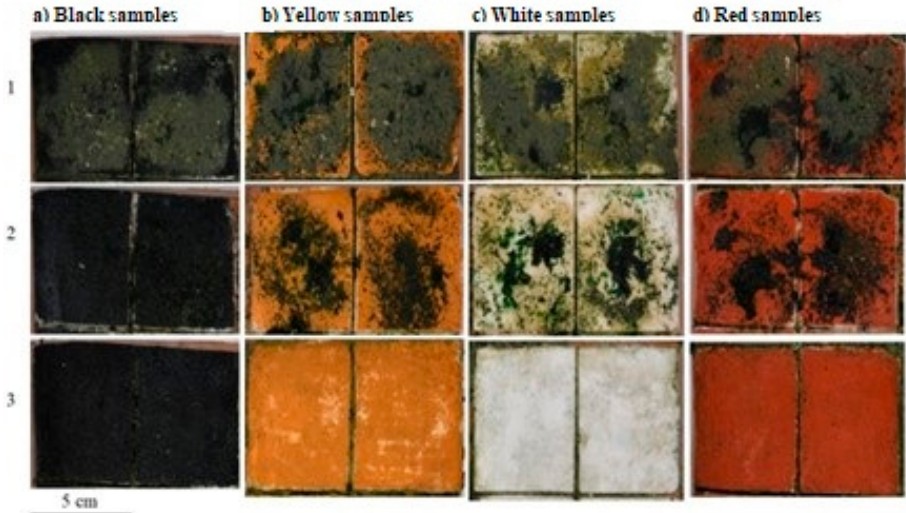

**Figure 5.** Samples were used for inoculation with a mixture of cyanobacteria and green algae for the UV-C treatments. (**a**) Black samples; (**b**) yellow samples; (**c**) white samples and (**d**) red samples. 1—before the treatments; 2—after the treatments; 3—24 h after the treatment.

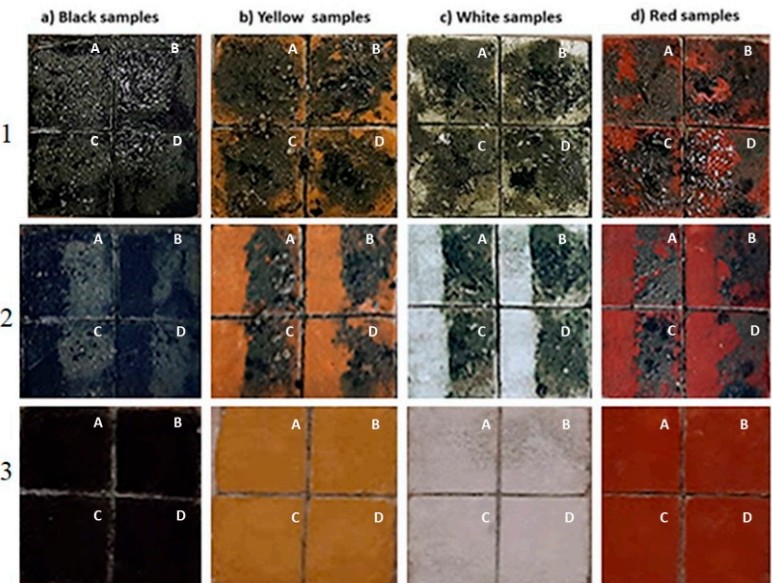

**Figure 6.** Samples treated with *ESSENZIO©* at different concentrations in demineralized water (10% (**A**), 20% (**B**), 50% (**C**), and pure (**D**)) and time of applications from 0 min (1) 30 min (2) to 90 min (3). (**a**) Black samples; (**b**) yellow samples; (**c**) white samples; and (**d**) red samples.

### 2.5. Absorption of Main Photosynthetic

Pigments following centrifugation (1′, 5000× *g*) and cell pigments were extracted using 2 mL of N,N-dimethylformamide (Spectrophotometer 7315, Jenway, Hong Kong, China), following the methodology illustrated by Carfagna et al. [12]. Chlorophyll content of cells was estimated spectrophotometrically (Spectrophotometer 7315, Jenway, Hong Kong, China). The absorbance at wavelengths 647 and 664 nm was used to determine the concentration of total chlorophyll as well as that of chlorophyll a and b, using the following equations:

$$\text{Chl Tot} = (A647 \times 17.9) + (A664 \times 8.08); \text{Chla} = (A664 \times 12.7) - (A647 \times 2.79);$$
$$\text{Chl b} = (A647 \times 20.78) - (A664 \times 4.88).$$

### 2.6. Colorimetric Analyses

Colorimetric investigations were conducted by light absorption in diffuse reflection using a MAYA 2000 pro (Ocean Insight, Oxford, UK) spectrophotometer in a Pump-Probe configuration. The light source used was a halogen lamp (CIE standard illuminant D65, Ocean Insight, Oxford, UK) with an emission spectrum in the VIS-NIR range 400–1000 nm. Before measurements, the device was calibrated using a white ceramic disk and a black trap portion. The commercial software OceanView 2.0 (Ocean Insight, Oxford, UK), with which the device was supplied, was used for the acquisition of the reflected spectra, the colorimetric data and for basic operations. Colorimetric parameters were calculated in the CIE L* a* b* 1976 colour space [13–15]. In this system, L* coordinate is lightness (L* = 0: black and L* = 100: white), a* coordinate represents the green/red component (negative values represent the green component and positive values represent the red component, through grey close to zero), and the b* coordinate refers to the blue/yellow component (negative values representing the blue component and positive values the yellow component, through grey close to zero). Samples with 4 different coloured areas—black, yellow, red and white—were investigated (Figures 5 and 6). The parameters L*, a* and b* were measured for the four colours in four different experimental conditions (M): before biofilm growth (M1), before biofilm growth but after the treatments considered (M2), after biofilm growth (M3), and after the treatments considered on biofilm (M4). For each colour, ten measurements in different points of the interested area were performed, and then the values were mediated to obtain the final parameters. Subsequently, for all treatments considered, the variations ΔL*, Δa* and Δb* were calculated considering the difference in measurements M4–M1, M4–M3 and M2–M1.

## 3. Results

### 3.1. XRF Analyses of the Pigments

The analysis of the red and yellow pigments shows the presence of calcium and iron, ascribable to a red ochre (probably hematite) and a yellow ochre (probably goethite) painted on a calcium carbonate plaster. In addition, the black pigment (as well as the white colour) revealed a peak only for the calcium, a sign of the organic nature of this colour, probably carbon black.

### 3.2. Effect of Essential Oil and UV-C on Artificial Biofilm and Control Test Samples

The samples completely covered by artificial biofilms consisting of several layers of intense green autotrophic microorganisms were subjected to the action of the essential oils of *ESSENZIO©* and the UV-C rays. The presence of an intense green biofilm indicates a high photosynthetic capacity.

After the treatments, the effects were clearly observable both on the samples treated with the oils and on those treated with UV-C; the biofilms grown on the exposed test samples were visibly bleached after the treatments (Figure 5). The progressive bleaching of the biofilms was related to the degradation of chlorophyll, as evidenced by spectrophotometric determinations.

The exposed test samples reported in Figures 5 and 6 show that all microorganisms were eradicated by both the essential oils and the UV-C treatment used in this study. A selection of test samples free from artificial biofilms, representative of each pigmentation, were then exposed to UV-C radiation. The colourimetric analysis performed on these samples did not show any chromatic alteration, proving that the treatment does not interfere with the support.

### 3.3. Measurement of Chlorophyll Concentration

The autotrophic microorganisms exposed to the different treatments showed a significant decrease in chlorophyll concentration. The decrease was more evident in the treatment with the essential-oils-based product (50% concentration in demineralized water, with a duration of application of 1 h and 30 min), compared to UV-C (irradiation conducted at

80 cm for a duration of 8 h a day every other day, for a total duration of twenty-four hours of exposure) (Figure 7). Coherently with the data for cell viability, these fluctuations were expressed as percentages.

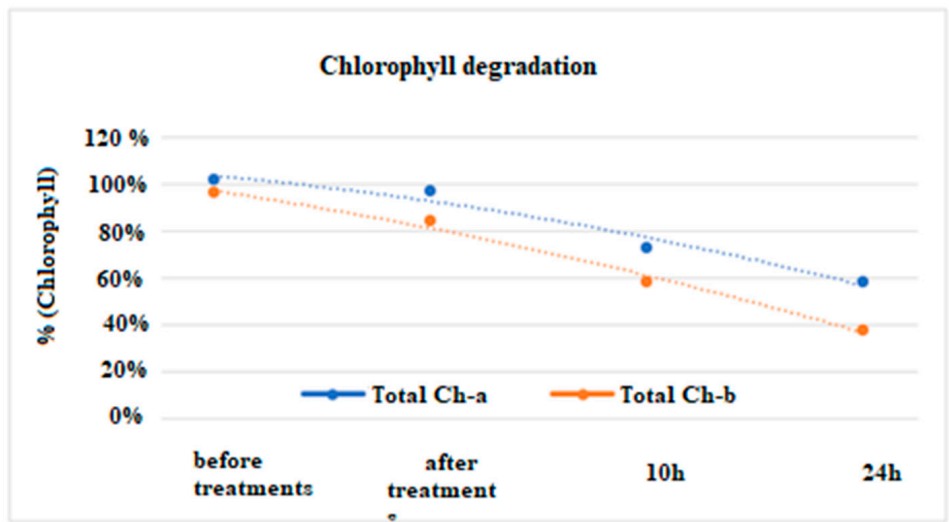

**Figure 7.** Total chlorophyll content in samples subjected to UV-C (ChlA) and *ESSENZIO©* (ChlB) treatment at different exposure times (before treatment, after treatment, 10 h, 24 h).

### 3.4. Colorimetric Analyses

Below, the colourimetric data for the black, yellow, red, and white colours of the samples treated with twelve different protocols based on the use of essential-oils-based products and with one protocol based on the use of UV-C radiation are reported. In particular, Tables 1 and 2 show the variations in the colourimetric parameters relating to the four colours of the samples achieved between measurements realized after the *ESSENZIO©* treatments on biofilm (M4) and, respectively, before the growth of the biofilm (M1) and after the growth of the biofilm (M3). In Tables 3 and 4, the same quantity referring to UV-C treatment are reported.

**Table 1.** Variations (M4–M1) in the colorimetric data of the black, yellow, red and white colours achieved between measurements realized on the samples after the twelve *ESSENZIO©* treatments considered on biofilm (M4) and before the growth of the biofilm (M1).

| Treatment: *ESSENZIO* | | Colour | | | | | | | | | | | |
| | | Black | | | Yellow | | | Red | | | White | | |
| Percentage in Water (%) | Time of Action (min) | ΔL* | Δa* | Δb* | ΔL* | Δa* | Δb* | ΔL* | Δa* | Δb* | ΔL* | Δa* | Δb* |
| 10 | 30 | 2.28 | 0.30 | 8.93 | 7.84 | 6.01 | 4.76 | 10.17 | −0.45 | 0.88 | −4.60 | −0.68 | 9.56 |
| 10 | 60 | 3.41 | −0.09 | 9.42 | 16.10 | 8.87 | 13.13 | 1.97 | −6.28 | 73.2 | −8.17 | −0.57 | 9.58 |
| 10 | 90 | 2.46 | −0.27 | −6.13 | 14.66 | 10.44 | 11.66 | 2.93 | −2.08 | −4.63 | −11.76 | −1.16 | 3.56 |
| 20 | 30 | 2.29 | −0.24 | −6.24 | 17.85 | 9.57 | 12.34 | 6.75 | 1.09 | −0.18 | −13.85 | 0.32 | 21.69 |
| 20 | 60 | 1.92 | −0.22 | −6.18 | 14.74 | 9.09 | 10.96 | 11.05 | 6.13 | 4.75 | −22.59 | −5.04 | 14.51 |
| 20 | 90 | 1.16 | −0.23 | −6.63 | 10.96 | 5.26 | 4.02 | 11.79 | 3.43 | 5.20 | −7.64 | −0.87 | 6.39 |
| 50 | 30 | 3.27 | −0.37 | −5.99 | 8.25 | 2.63 | 1.95 | 4.36 | −5.83 | −6.64 | −13.64 | −0.46 | 6.16 |
| 50 | 60 | 4.24 | −0.1 | −5.51 | 9.21 | 2.65 | 2.75 | 6.65 | −1.53 | −2.61 | −13.95 | −0.57 | 5.76 |
| 50 | 90 | 2.00 | −0.29 | −6.79 | 16.41 | 5.46 | 11.36 | 4.35 | −2.24 | −3.01 | −2.58 | −0.97 | 6.06 |
| 100 (pure) | 30 | 1.40 | −0.42 | −6.57 | 13.90 | 5.72 | 8.36 | 4.33 | −4.03 | −6.40 | −10.08 | −0.54 | 6.45 |
| 100 (pure) | 60 | 1.69 | −0.27 | −6.94 | 23.48 | 12.58 | 19.86 | 2.95 | −3.11 | −5.08 | −3.12 | −1.47 | 3.42 |
| 100 (pure) | 90 | 2.87 | −0.39 | −7.32 | 18.75 | 7.26 | 11.72 | 7.78 | 0.98 | −1.51 | −12.10 | −0.69 | 6.99 |

**Table 2.** Variations (M4–M3) in the colorimetric data of the black, yellow, red and white colours achieved between measurements realized on the samples after the twelve *ESSENZIO©* treatments considered on biofilm (M4) and after the growth of the biofilm (M3).

| Treatment: *ESSENZIO* | | Colour | | | | | | | | | | | |
| --- | --- | --- | --- | --- | --- | --- | --- | --- | --- | --- | --- | --- | --- |
| | | Black | | | Yellow | | | Red | | | White | | |
| Percentage in Water (%) | Time of Action (min) | ΔL* | Δa* | Δb* | ΔL* | Δa* | Δb* | ΔL* | Δa* | Δb* | ΔL* | Δa* | Δb* |
| 10 | 30 | −16.31 | 3.34 | −9.28 | 11.49 | 23.55 | 19.66 | 7.24 | 33.23 | 7.73 | 47.95 | 2.55 | −15.64 |
| 10 | 60 | −15.18 | 2.95 | −8.79 | 19.75 | 26.41 | 28.03 | 1.02 | 27.40 | 80.05 | 44.38 | 2.66 | −15.62 |
| 10 | 90 | −16.13 | 2.77 | −24.34 | 18.31 | 27.98 | 26.56 | 0 | 31.60 | 2.22 | 40.79 | 2.07 | −21.64 |
| 20 | 30 | −16.30 | 2.80 | −24.45 | 21.5 | 27.11 | 27.24 | 3.82 | 34.77 | 6.67 | 38.70 | 3.55 | −3.51 |
| 20 | 60 | −16.67 | 2.82 | −24.39 | 18.39 | 26.63 | 25.86 | 8.12 | 39.81 | 11.60 | 29.96 | −1.81 | −10.69 |
| 20 | 90 | −17.43 | 2.81 | −24.84 | 14.61 | 22.80 | 18.92 | 8.86 | 37.11 | 12.05 | 44.91 | 2.36 | −18.81 |
| 50 | 30 | −15.32 | 2.67 | −24.20 | 11.90 | 20.17 | 16.85 | 1.43 | 27.85 | 0.21 | 38.91 | 2.77 | −19.04 |
| 50 | 60 | −14.35 | 2.94 | −23.72 | 12.86 | 20.19 | 17.65 | 3.7 | 32.15 | 4.24 | 38.60 | 2.66 | −19.44 |
| 50 | 90 | −16.59 | 2.75 | −25.00 | 20.06 | 23.00 | 26.26 | 1.42 | 31.44 | 3.84 | 49.97 | 2.26 | −19.14 |
| 100 (pure) | 30 | −17.19 | 2.62 | −24.78 | 17.55 | 23.26 | 23.26 | 1.4 | 29.65 | 0.45 | 42.47 | 2.69 | −18.75 |
| 100 (pure) | 60 | −19.90 | 2.77 | −25.15 | 27.13 | 30.12 | 34.76 | 0.02 | 30.57 | 1.77 | 49.43 | 1.76 | −21.78 |
| 100 (pure) | 90 | −15.72 | 2.65 | −25.53 | 22.40 | 24.80 | 26.62 | 4.85 | 34.66 | 5.34 | 40.45 | 2.54 | −18.21 |

**Table 3.** Variations (M4–M1) in the colorimetric data of the black, yellow, red and white colours achieved between measurements realized on the samples after the UV-C radiation treatments on biofilm (M4) and before the growth of the biofilm (M1).

| Treatment: UV-C Radiation | | | |
| --- | --- | --- | --- |
| Colour | ΔL* | Δa* | Δb* |
| Black | −0.72 | 0.12 | 0.86 |
| Yellow | 3.12 | 3.32 | 1.81 |
| Red | −1.53 | −7.28 | −6.04 |
| White | −2.42 | 2.72 | 10.34 |

**Table 4.** Variations (M4–M3) in the colorimetric data of the black, yellow, red and white colours achieved between measurements realized on the samples after the UV-C radiation treatments on biofilm (M4) and after the growth of the biofilm (M3).

| Treatment: UV-C Radiation | | | |
| --- | --- | --- | --- |
| Colour | ΔL* | Δa* | Δb* |
| Black | −19.31 | −3.34 | −14.91 |
| Yellow | 6.77 | 20.29 | 16.71 |
| Red | −4.46 | 26.40 | 0.81 |
| White | 50.15 | 5.95 | −14.86 |

In Figure 8, for each colour, the reflectance spectrum measured before the biological growth (red lines) and after the contamination with the subsequent biocidal treatments (black lines) of the samples treated with UV-C radiation are reported.

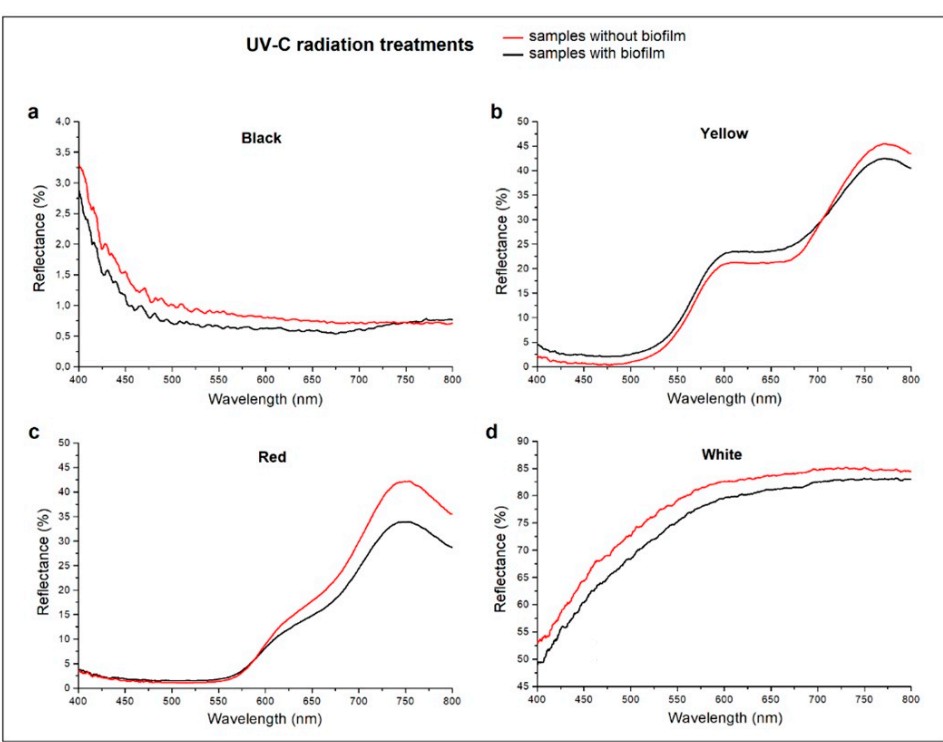

**Figure 8.** Reflectance spectra measured for samples exposed to UV-C radiation before biofilm growth (red lines) and after the contamination with the subsequent biocidal treatments (black lines) referring to the colours (**a**) black, (**b**) yellow, (**c**) red and (**d**) white.

## 4. Discussion

The extreme microclimatic conditions of the archaeological sites led to the proliferation of different biological species stimulated by the formation of a polysaccharide matrix that contributes to the genesis of the biofilm [1,16–22]. In our study, we observed the development of different organisms on the wall paintings present in room SB-E0-R07 of the Archeological Park of Baia. In this study, we propose two alternative methods to reduce the presence of biodeteriogens on the painted surfaces: essential-oils-based products and UV-C. Our experiments show that both these methods are effective in the reduction of biodeteriogens under laboratory conditions. In detail, tests were performed using the thyme and oregano essential-oils-based product *ESSENZIO©*; in particular, the product was applied by brush at different percentages of dilution (10%, 20%, 50% in demineralized water and pure) and left on for different durations of action (30 min, 1 h and 1 h and 30 min). Both of these essential oils contain phenols (respectively, thymol and carvacrol), chemical molecules known for their antimicrobial action due to the presence of oxidrilic groups. These chemical groups could cause an alteration in the enzymatic reactions of the biological species, inhibiting their proliferation [23–26]. Due to their antimicrobial proprieties, Origanum vulgare and Thymus vulgaris essential oils were recently used to control biological growth on artistic manufacts, highlighting their effectiveness in conservation strategies [23,27–30], as shown in the study carried out in the Vatican Gardens by Devreux et al. [9], on stone walls in Persepolis and Salluzzo by Favero Longo et al. [31], and by the experiments on some microbial strains isolated from different cultural objects executed by Rotolo et al. [29].

Based on the results obtained through the experiments carried out, the procedures have shown the greater effectiveness of *ESSENZIO©*, particularly at a dilution of 50% in demineralized water with a duration of application of 1 h and 30 min.

The colorimetric analysis confirmed the success of the treatments performed on the samples, with different results for the pigments investigated, as shown in Tables 1 and 2.

The black samples treated with *ESSENZIO©* show, in all the procedures of application, a slight whitening of the testers due to a thinning of the painted layer and incomplete carbonation at the moment of the experimentation and during the removal of the product, in agreement with modest increases in L\* values between 1.16 and 4.34 (Table 1). The negative rates of the L\* achieved comparing the values in the presence of the biofilm and after the treatments are indicative, on the other hand, of a good reestablishment of the luminosity of the tester after the treatments, confirming the success of the procedures, ulteriorly validated by the increase in the Δa\* and decrease in the Δb\* parameters (Table 2).

The yellow samples treated with *ESSENZIO©* show, as in the case of the black testers, an increase in the L\* parameters between 7.84 and 23.48 (Table 1) caused, once again, by a thinning of the painted layer after the rinsing operations of the product. The increase in the Δa\* and Δb\* parameters (Tables 1 and 2), instead, indicate the effective restoration of the original conditions of the pigment, with a slight colour change caused by the proliferation of the biological species that was not visible to the eye.

The red testers indicated an increase in the L\* values between 1.97 and 11.79 (Table 1), in agreement with what was said earlier. The positive tendency of this parameter confirms the success of the treatments in the removal of the biofilm. This aspect is confirmed also by the increase in the Δa\* and Δb\* parameters (Table 2) which show, however, slight variations in the original values (Table 1) due to a modest alteration of the original colour caused by the biodeteriogens.

The white samples, instead, indicate a decrease in the L\* parameters, a sign of the darkening of the surface visible to the naked eye, particularly in the testers treated with *ESSENZIO©* diluted at 10% and 20% in demineralized water (Table 1). However, the positive values of this parameter in the range 29.96 and 49.97 compared to the measurements taken on the contaminated samples show the general success of the treatments (Table 2).

The presence of visible green spots in the frame treated with *ESSENZIO©* diluted at 10% and 20% on the samples is indicated also by the decrease in the Δa\* and increase in the Δb\* values compared to the original parameters, signs of a colour change in the green and the yellow regions (Table 1). However, the increase in the Δa\* and increase Δb\* after the treatments indicate a good outcome of the procedures applied (Table 2).

Concerning the UV-C, the samples were exposed to irradiation at a distance of 80 cm for a duration of 8 h a day every day, for a total duration of 24 h of exposure. Prolonged exposure to radiation causes damage to the biological organisms due to the high absorbance of their proteins [32,33]. Over the last decades, the absence of any adverse effect on operators, environment and cultural heritage, combined with the effectiveness of this method, has led to the application of UV-C rays to the conservation procedures [16,22,34]; this was demonstrated, in particular, in the Cennamo et al. study conducted on Necropoli of Porta Nocera (Pompei) wall paintings [7] and by the application of this procedure in Moidons Caves (France) carried out by Borderie et al. [32,35,36], showing encouraging results.

The black samples treated with UV-C at a distance of 80 cm showed a huge decrease in L\* (Table 4), with a slight increase compared to the initial value (Table 3) due to the colour change caused by the biodeteriogens. The Δa\* and Δb\* values (Table 3), on the other hand, indicate a small increment, confirming the release of pigments by the biological species. However, the decrease in these parameters in relation to the contaminated surfaces indicate the success of the biocidal treatment (Table 4). The reflectance spectrum (Figure 8a) shows a weak reflectance reduction in the sample after the UV-C treatment, in agreement with the mild modification of the colour caused by the biodeteriogens.

The yellow samples show an increase in the ΔL\* compared to the initial value, referable, once again, to the mechanical removal of the biological species after the treatment. This aspect is confirmed also by the slight increment of the reflectance of the sample after the UV-C treatment, as visible in Figure 8b. The positive value of the ΔL\*, however, indicate the success of the biocidal method. This observation is further confirmed by the increase in the Δa\* and Δb\* parameters after the application of the biocidal protocol (Tables 3 and 4).

The red sample indicates a slight decrease in $\Delta$L* equal to $-1.53$ due to a darkening of the original pigment visible to the naked eye caused by the proliferation of the artificial biofilm. This observation is also confirmed by the decrease in the $\Delta$a* and $\Delta$b* parameters measured with respect to the untreated sample (Table 3), which indicates a colour change of the pigment towards the green and yellow regions. This aspect is confirmed also by the decrease in the reflectance of the sample after the UV-C treatment, as visible in the reflectance spectrum in Figure 8c. However, the increase in these two parameters shown in Table 4 (especially for $\Delta$a*) indicate the effectiveness of the biocidal treatment, with a good restoration of the original proprieties of the painted layer.

Finally, the white sample exposed to UV-C radiation indicated a decrease in the $\Delta$L* equal to $-2.42$ due to a darkening of the surface caused by the disfiguring action of the biodeteriogens, a consequence also of the increase in the $\Delta$a* and $\Delta$b* (Table 3). The reflectance spectrum in Figure 8d confirms this data, with a decrease in the curve after the biocidal treatment. However, comparing the $\Delta$L* values shown in Tables 3 and 4, it is possible to notice a huge increase when it is calculated in the presence of the biofilm, demonstrating the effectiveness of the treatment. This observation is further confirmed by the increase and decrease in $\Delta$a* and $\Delta$b*, respectively, compared to the values recorded before the biocidal treatment (Table 4).

In reference to the graphs in Figure 8, it should be noted that in the case of the black (Figure 8a), red (Figure 8c) and white (Figure 8b) colours, the reflectance measured after treatment (black line) is on average slightly lower on the whole spectrum than that measured before biological growth (red line). These data indicate a slight increase in the opacity of these colours due to the combined action of the biological film and the consequent UV treatment carried out. Differently, in the case of the yellow colour (Figure 8b), the reflectance measured after treatment (black line) is slightly higher than that measured before (red line) up to about a wavelength of 700 nm, while after this value, it is lower, which is in agreement with what was found with the other colours. This last achievement highlights how the effects on this colour of both the growth of the biofilm and the treatment performed affect the final state of the surface and are considered lower.

Colorimetric analysis was also used to investigate the effects of the treatments with *ESSENZIO* and UV-C radiation on samples without biofilm. Parameters $\Delta$L*, $\Delta$a* and $\Delta$b* were calculated by considering the difference of the measurements performed before biofilm growth (M1) and before biofilm growth but after the treatments considered (M2). The results obtained show small variations in parameters L*, a* and b* for both approaches, which in each case were less than 0.03; therefore, they show the absence of substantial chromatic variations in the pigments caused by the tested protocols.

## 5. Conclusions

The results of the experimentations carried out during the present study successfully demonstrated the effectiveness of UV-C irradiation and *ESSENZIO©* as alternative methods to traditional biocides for the reduction of biological growth on artistic artifacts. In particular, the following conclusions have been drawn:

1. The protocol conducted with *ESSENZIO©* at a dilution of 50% in demineralized water with a duration of application of 1 hour and 30 min showed the best results for the removal of the artificial biofilm on the painted samples, despite not demonstrating any biocidal power—only a cleaning action. The Origanum vulgare and Thymus vulgaris essential oils, the main ingredients of *ESSENZIO©*, contain, respectively, thymol and carvacrol, phenolic compounds able to inhibit the enzymatic reactions of the biological species due to the acidic nature of hydroxyls within in the aromatic groups [26].

2. UV-C treatments, conducted at a distance of 80 cm and with a wavelength of $\lambda$ max = 254 nm and a duration of exposure of 8 h, proved to be very useful in countering the proliferation of biological species. As demonstrated by previous studies [7], UV-C radiation plays an important role in the degradation of the proteins that make up the two photosystems and some of the enzymes involved in photosynthesis [18]. The alterations

caused by UV-C on photosynthetic microorganisms are described as "whitening" due to the evident reduction in the chromatic intensity of the pigments [19]; a similar result was also obtained with the use of *ESSENZIO©*.

3. The colorimetric analysis of the samples treated with essential-oils-based products indicated an increase in the L* parameter after the biocidal protocol compared to the values registered on the painted surfaces contaminated with artificial biofilm. These data confirm the effectiveness of the treatments. The slight variations in the Δa* and Δb* parameters, on the other hand, indicate a mini-mum colour change caused mainly by the proliferation of the biological species, with a peak in the values registered on the testers treated with *ESSENZIO©* diluted at 10% and 20% in demineralized water, a sign of incomplete removal of biodeteriogens.

4. The colorimetric investigations on the samples treated with UV-C showed an increase in the L* parameter after the biocidal protocol, confirming the success of the proce-dures and the restoration of the optical proprieties of the testers before the contamination with artificial biofilm. Instead, the variation in the Δa* and Δb* parameters indicates a slight variation in the original colour of the surfaces due to the proliferation of the artificial biofilm. This aspect is further confirmed by the increase in the reflectance spectra after the application of the biocidal protocol (Figure 8).

5. Spectrophotometric readings indicate a major decrease in the chlorophyll concentra-tion in the samples treated with *ESSENZIO©* (50% concentration in demineralized water with a time of application of 1 h and 30 min), compared to UV-C (irradiation conducted at 80 cm for a duration of 8 h a day every other day, for a total duration of 24 h of expo-sure). These data demonstrate the higher effectiveness of the essential-oils-based product compared to UV-C irradiation for the treatment of the artificial biofilm.

6. The comparison of the two methods and the results of the subsequent diagnostic investigations have led, in consequence, to suggest a combined use of *ESSENZIO©*, efficient in the first removal of the biofilms, and UV-C rays, ideal for preventing the biological proliferation of the biodeteriogens. This combined protocol could be a valid alternative to chemical biocides, which are highly toxic for operators and the environment, guaranteeing, at the same time, the control of biodeteriogen diffusion on artistic crafts without any side effects for the restorers and the ambiance.

**Author Contributions:** Conceptualization, P.C. and G.T.; methodology, R.S., E.C. and M.R.; software, M.R. and S.C.; validation, P.C., R.S. and G.T.; formal analysis, R.S., G.T. and M.R.; investigation, R.S.; resources, P.C. and G.T.; data curation, P.C. and G.T.; writing—original draft preparation, R.S., P.C. and G.T.; writing—review and editing, P.C., R.S. and G.T.; visualization P.C., R.S. and G.T.; supervision, P.C.; project administration, P.C. and G.T.; funding acquisition, P.C. All authors have read and agreed to the published version of the manuscript.

**Funding:** This research was funded by PNRR_PE CHANGES-Cultural Heritage Active Innovation for Sustainabel Society. SPOKE n 6 –"History, conservation and restoration of cultural heritage".

**Institutional Review Board Statement:** Not applicable.

**Informed Consent Statement:** Not applicable.

**Data Availability Statement:** Not applicable.

**Conflicts of Interest:** The authors declare no conflict of interest.

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
