# Peer review of "UV-C Irradiation and Essential-Oils-Based Product as Tools to Reduce Biodeteriorates on the Wall Paints of the Archeological Site of Baia (Italy)"

_coatings, doi:10.3390/coatings13061034_

Round 1
Reviewer 1 Report
The combined action of UV-based irradiation and essential oils appears to be a standard method for the removal of bacteria, biofilms and algae from surfaces. In general, there is no comparison with the literature or an adequate discussion about the mechanisms for interaction of UV-C radiation and essential oils with surfaces. A discussion about the explored essential oil is minimally necessary. More references are necessary to validate the discussion.
Additional points
I suggest substitute the tile to “UV-C IRRADIATION AND ESSENTIAL OILS BASED PROD-
UCT AS TOOLS TO REDUCE BIODETERIORATES ON THE
WALL PAINTS OF THE ARCHEOLOGICAL SITE OF BAIA (Italy)”
Please revise he Figure captions, avoiding the use of “Graphic of the”
Form an image it is not possible to conclude that “The exposed test samples, presented in Figures 5 and 6, showed that all microorganisms were eradicated by oils essential, and the UV-C intensity used in this study”
The response in Fig. 8 for reflectance is poorly explored. Please revise it.
Based on these comments, I consider that manuscript is not ready for acceptance and required a profound a revision before a new submission.
A revision from a native is necessary.
Author Response
Thanks for the review, we have answered your questions and we have deepened the topics suggested above all in the discussion
Revisione 1
The combined action of UV-based irradiation and essential oils appears to be a standard method for the removal of bacteria, biofilms and algae from surfaces. In general, there is no comparison with the literature or an adequate discussion about the mechanisms for interaction of UV-C radiation and essential oils with surfaces. A discussion about the explored essential oil is minimally necessary. More references are necessary to validate the discussion.
Additional points
I suggest substitute the tile to “UV-C IRRADIATION AND ESSENTIAL OILS BASED PROD-
UCT AS TOOLS TO REDUCE BIODETERIORATES ON THE
WALL PAINTS OF THE ARCHEOLOGICAL SITE OF BAIA (Italy)”
R: done
Please revise he Figure captions, avoiding the use of “Graphic of the”
R: done
Form an image it is not possible to conclude that “The exposed test samples, presented in Figures 5 and 6, showed that all microorganisms were eradicated by oils essential, and the UV-C intensity used in this study”
R: we changed to: The exposed test samples, reported in figures 5 and 6, show that all microorganisms were eradicated by both the essential oils and UV-C treatment used in this study.
The response in Fig. 8 for reflectance is poorly explored. Please revise it.
R: We thank the Reviewer for his suggestions. We have added some text to the manuscript explaining and commenting better the graphs of Fig.8.
Based on these comments, I consider that manuscript is not ready for acceptance and required a profound a revision before a new submission.
The manuscript has been revised for the english language

Reviewer 2 Report
In this manuscript, Paola et al reported the UV-C irradiation and essential iols based product as tools to reduce biodteriorates on the wall paints of the archeological site of baia,the paper can be accepted after the following issue were concerned.
1. The scale bar is missing in Fig. 5 and Fig. 6
2. The conclusions need to be expanded, the proposal should be highlighted.
N/A
Author Response
The manuscript has been revised for the english language
Comments and Suggestions for Authors
In this manuscript, Paola et al reported the UV-C irradiation and essential iols based product as tools to reduce biodteriorates on the wall paints of the archeological site of baia,the paper can be accepted after the following issue were concerned.
- The scale bar is missing in Fig. 5 and Fig. 6
R: have been added
- The conclusions need to be expanded, the proposal should be highlighted.
R: conclusions have been rewritten

Reviewer 3 Report
This study seems interesting but inappropriately carried out. The methodology needs to be well written, the results should be discussed and compared with other relevant literature.
- Rephrase the first sentence of the abstract
- Check line 29, “ad”
- Overall the abstract is not informative; lacks relevant information at a glance. You should had more specifics in terms of microorganism sp.; essential oil used. Additionally, information on results i.e. dose range and effective dose. Etc.]
- Please rewrite the method in section 2.2 for better clarity.
- L125 Origan essential oil? Check, please
- How was oil dilution achieved by demineralized water>
- Fig 3; value of axis can be in 1 에
- Fig 5 caption is unclear. Also no row labels. Which samples are on the 2nd and 3rd rows?
- The discussion should be improved with relevant references.
- Please rewrite the conclusion, at the moment it doesn’t read like a concluding section of a research article.
Please check detailedly for grammatical errors and omissions. Also, some sentences are unclear.
Author Response
This study seems interesting but inappropriately carried out. The methodology needs to be well written, the results should be discussed and compared with other relevant literature.
- Rephrase the first sentence of the abstract
R: done
- Check line 29, “ad”
R: done
- Overall the abstract is not informative; lacks relevant information at a glance. You should had more specifics in terms of microorganism sp.; essential oil used. Additionally, information on results i.e. dose range and effective dose. Etc.]
- Please rewrite the method in section 2.2 for better clarity.
R: done
- L125 Origan essential oil? Check, please
R: done
- How was oil dilution achieved by demineralized water>
R: we add in the manuscript
- Fig 3; value of axis can be in 1 에
R: done
- Fig 5 caption is unclear. Also no row labels. Which samples are on the 2nd and 3rd rows?
R: the caption has been rewritten
- The discussion should be improved with relevant references.
R: conclusions have been rewritten
- Please rewrite the conclusion, at the moment it doesn’t read like a concluding section of a research article.
R: conclusions have been rewritten
The manuscript has been revised for the english language

Reviewer 4 Report
In this paper, UV-C irradiation and essential oil-based products were used to remove biofilm, and the results showed that both methods were effective in removing biofilm. However, there are some important points which should be addressed before a further consideration. I suggest major revision of the manuscript based on the following comments:
1. In lines 122-123, "(a) white samples; (b) red samples; (c) yellow samples and black samples (d)" describe sample colors inconsistent with those in Fig. 5 and Fig. 6 (Figure 5,6). Please check carefully.
2. In lines 128 and 134, it is stated that “(Figures 5)” and “(Figure 6)” are the “essential oil-based product treatments” and the “UV-C treatments”, respectively, but the actual Fig. 5 is the UV-C treatments, while Fig. 6 is the essential oil-based product treatments. Please adjust the order of the two figures.
3. Suggested addition of spectrograms for XRF analysis
4. Suggestions to improve the clarity of Fig. 6.
5. Please indicate the treatment conditions, such as time or concentration, for the samples in the first, second and third rows in Fig. 5 and Fig. 6, respectively.
6. In line 291, "The reflectance spectrum (Figure 11a) shows a weak reflectance reduction... ", there is no Fig. 11 in the article, please double check and revise.
7. Please keep the formatting of the sequence symbols in the references consistent.
8. Please revise the language of the article to be as concise and clear as possible.
The English language should be modified to be as concise and clear as possible.
Author Response
In this paper, UV-C irradiation and essential oil-based products were used to remove biofilm, and the results showed that both methods were effective in removing biofilm. However, there are some important points which should be addressed before a further consideration. I suggest major revision of the manuscript based on the following comments:
- In lines 122-123, "(a) white samples; (b) red samples; (c) yellow samples and black samples (d)" describe sample colors inconsistent with those in Fig. 5 and Fig. 6 (Figure 5,6). Please check carefully.
R: done
- In lines 128 and 134, it is stated that “(Figures 5)” and “(Figure 6)” are the “essential oil-based product treatments” and the “UV-C treatments”, respectively, but the actual Fig. 5 is the UV-C treatments, while Fig. 6 is the essential oil-based product treatments. Please adjust the order of the two figures.
R: done
- Suggested addition of spectrograms for XRF analysis
R: We think they are unnecessary as we have entered the results.
- Suggestions to improve the clarity of Fig. 6.
R: done
- Please indicate the treatment conditions, such as time or concentration, for the samples in the first, second and third rows in Fig. 5 and Fig. 6, respectively.
R: done
- In line 291, "The reflectance spectrum (Figure 11a) shows a weak reflectance reduction... ", there is no Fig. 11 in the article, please double check and revise.
R: done
- Please keep the formatting of the sequence symbols in the references consistent.
R: done
- Please revise the language of the article to be as concise and clear as possible.
R: done. All the unclear parts have been rewritten and the manuscript has been revised for the language

Round 2
Reviewer 1 Report
In view of the modifications provided by the authors, I consider that manuscript can be accepted as is.
Author Response
Dear Reviewer,
Thanks again for your suggestions
Paola Cennamo
Reviewer 3 Report
- Some corrections were not attended to appropriately. Please try to state the corrections in detail instead of commenting “done.”
- Conclusion is unnecessarily too long and thus not informative. It seems more like a rewritten discussion
- Please explain how demineralized water achieved oil dilution since Oil and water do not mix.
- The discussion still lacks relevant references. The authors responded to my last review comments by attaching 6 to 2 sentences (L 251 & 293). This doesn't seem right. I instead suggested that you compare the results of this study with other “relevant studies. Please correct it appropriately.
- Fig. 3 & Fig. 7, Y-axis needs to be formatted to 1 or 2 decimal places.
Author Response
Thank the reviewer for his comments and apologize for the previous replies where we put only done. We also insert the answers of the 1st revision.
- Some corrections were not attended to appropriately. Please try to state the corrections in detail instead of commenting “done.”
-
- Rephrase the first sentence of the abstract
R: The phrases to the lines 15-16have been inserted: This study is aimed to compare, through laboratory experimentations, the efficiency of UV-C irradiation and an essential oils-based product as tools to reduce the biofilm identified in a semi-hypogeum room located in the archeological park of Baia, Italy. Lines 29-30:Results showed that biofilms were successfully removed by irradiation of the surfaces and by the essential oils-based product at a dilution of 50% in demineralized water with a time of application of one hour and 30 minutes; in addition,
- Overall the abstract is not informative; lacks relevant information at a glance. You should had more specifics in terms of microorganism sp.; essential oil used. Additionally, information on results i.e. dose range and effective dose. Etc.]
- Please rewrite the method in section 2.2 for better clarity.
R: lines118-120 have been rewritten
- Please explain how demineralized water achieved oil dilution since Oil and water do not mix.
- R:
In the formulation of ESSENZIO© there are mainly Thyme and Origan essential oils but note is the presence of other chemical compounds (e.g. surfactants). These allow the dilution in demineralized water of the product without any division of phase between the two compounds.
- The discussion still lacks relevant references. The authors responded to my last review comments by attaching 6 to 2 sentences (L 251 & 293). This doesn't seem right. I instead suggested that you compare the results of this study with other “relevant studies. Please correct it appropriately.
R: WE add lines 252-255
as shown in the study carried out in the Vatican Gardens by Devreux et al., [9] on stone walls in Persepolis and Salluzzo by Favero Longo et al. [30] and by the experimentations on some microbial strains isolated from different cultural objects executed by Rotolo et al. [33] .
Fig. 3 & Fig. 7, Y-axis needs to be formatted to 1 or 2 decimal places.
R: we we formatted it with 2 decimal places.
add lines 297-299:
as shown, in particular, in Cennamo et al. study conducted on Necropoli of Porta Nocera (Pompei) wall paintings [7] and in the application of this procedure in Moidons Caves (France) carried out by Borderie et al. [27, 35, 36], showing encouraging results.
The conclusions have been rewritten in order to respond to the requests of the other reviewers

Reviewer 4 Report
Please check the full text carefully, there are many "。" causing the article garbled.
Moderate editing of English language
Author Response
Dear Reviewer,
thank you for your suggestions
best regards
Paola Cennamo